# Measuring and addressing the childhood tuberculosis reporting gaps in Pakistan: The first ever national inventory study among children

**Razia Fatima[1], Aashifa Yaqoob[1,2]\*, Ejaz Qadeer[3], Sven Gudmund Hinderaker[2], Aamer Ikram[1,4], Charalambos Sismanidis[5]**

**1** Common Management Unit (HIV/AIDS, TB & Malaria), Islamabad, Pakistan, **2** University of Bergen, Bergen, Norway, **3** Pakistan Institute of Medical Sciences, Islamabad, Pakistan, **4** National Institute of Health, Islamabad, Pakistan, **5** Global TB Programme, World Health Organization, Geneva, Switzerland

\* aashifa.yaqoob@gmail.com

**Data Availability Statement:** Some data sharing restrictions are imposed by the program. Special permission for program is required for data

## Abstract

### Introduction

Tuberculosis in children may be difficult to diagnose and is often not reported to routine surveillance systems. Understanding and addressing the tuberculosis (TB) case detection and reporting gaps strengthens national routine TB surveillance systems.

### Objective

The present study aimed to measure the percentage of childhood TB cases that are diagnosed but not reported to the national surveillance system in Pakistan.

### Design

The study design was cross sectional. The study was nationwide in 12 selected districts across Pakistan, each representing a cluster. Health facilities that diagnose and treat childhood TB from all sectors were mapped and invited to participate. Lists of child TB cases were created for the study period (April-June 2016) from all study facilities and compared against the list of child TB cases notified to the national TB surveillance system for the same districts and the same period.

### Results

All public and private health facilities were mapped across 12 sampled districts in Pakistan and those providing health services to child TB cases were included in the study. From all private health facilities, 7,125 children were found with presumptive TB during the study period. Of them, 5,258 were diagnosed with tuberculosis: 11% were bacteriologically-confirmed and 89% clinically-diagnosed; only 4% were notified to National TB Control Program. An additional 1,267 children with TB were also registered in the National TB Control Program. Underreporting was measured to be 78%.

availability request because of office policy restriction for data sharing. Data can be accessed by contacting IRC Ethics Committee [irb@ntp.gov.pk], Common Management unit (HIV/AIDS, TB & Malaria) to which data requests can be sent.

**Funding:** The study was carried out with funding from UNITAID, channeled through the STEPTB project that was led by the TB Alliance and WHO. The funders had no role in study design, data collection and analysis, decision to publish, or preparation of the manuscript.

**Competing interests:** The authors have declared that no competing interests exist.

## Conclusion

This is the first nationwide childhood TB inventory study globally and confirmed that childhood TB underreporting is very high in Pakistan. TB surveillance in the country must be strengthened to address this, with particular attention to guiding and supporting general practitioners and pediatricians to notify their TB cases.

## Introduction

Every day globally up to 650 children (aged <15 years) lose their lives due to tuberculosis (TB), a preventable and curable disease [1]; children account for 10% of the new TB cases notified in 2017 [2]. In countries worldwide, the reported percentage of all TB cases occurring in children varies from 3% to more than 25% [3]. However, the estimates of child tuberculosis cases have been unreliable partly because of limited surveillance data of children in many countries; also very few epidemiological studies are done [4]. TB in children is often missed or overlooked due to non-specific symptoms and difficulties in diagnosis; this has made it difficult to assess the actual magnitude of the burden, [5]. Understanding and addressing the TB case detection and reporting gaps may strengthen national routine TB surveillance systems and allow for accurate monitoring of progress with TB prevention and care targets set by National Tuberculosis Programmes and their partners. An inventory study measuring the level of underreporting of child TB may show the gap between diagnosis and reporting of child TB cases [6].

During 2006–07, NTP Pakistan developed its national childhood TB policy guidelines in collaboration with Pakistan Pediatric Association (PPA) [7]. These guidelines for TB in children, including a scoring chart for helping diagnosis, were among the first of its kind in developing countries. The purpose of the guidelines was to help the pediatricians, physicians and other health workers to improve and standardize diagnosis and case management among children with TB. Currently, the PPA scoring chart is shared with all pediatricians to diagnose TB in children.

In Pakistan, TB accounts for 8–20% of all deaths in children [8]. According to TB notification 2018 in Pakistan, out of around 369,000 TB cases 13% were children. The private health sector in Pakistan is very big: 85% of patients choose to seek care in the private sector which is large and unregulated and is comprised of both qualified and unqualified service providers in the disciplines of allopathy, homeopathy and tibb (Traditional Herbal Medicine) [9]. TB drugs are available in private pharmacies and there are many large private teaching hospitals present in the country that are managing infectious and non-infectious diseases. In 2017, 28% of the TB notification in Pakistan was reported by private sector, [10] but the proportion of general practitioners (GPs) and paediatrician reporting is quite low. Therefore, it is expected that the national surveillance system is missing many child TB cases from the private sector. An inventory study carried out in 2012 showed that 27% of adult TB cases were not notified to the National Tuberculosis Program (NTP) in Pakistan [11]. Unpublished data from that study showed that underreporting was 2.5 times higher in age group < 15 years. One could hypothesize that the percentage of children could be even higher, since the study did not specifically include in its sampling frame facilities/private practitioners who would also diagnose TB in children. Therefore, the current study was done to assess the level of underreporting of child TB in Pakistan.

## Materials and methods

### Design

This was a nationwide cluster-based cross sectional study.

### Study area

This study was a national cluster-based, prospective study targeting all facilities in the selected study areas that diagnose and treat childhood TB in Pakistan. To ensure national representation, at least one district was selected from each of the 6 provinces/regions, while the number of the remaining 6 districts were split proportional to province/region population size of children (Table 1). In these study areas we collected data on cases diagnosed by all health-care providers within these areas for a specified time period, followed by record-linkage to the electronic, case-based NTP database[6].

Districts were defined to be the study clusters and were randomly sampled from all four provinces (Balochistan, Khyber Pakhtunkhwa, Punjab and Sindh) and two regions (Gilgit Baltistan and Azad Jammu & Kashmir). All eligible facilities from sampled districts were enumerated and invited to participate in the study. Federally Administered Tribal Area districts were excluded from the sampling frame due to security concerns.

### Sample size considerations

The required sample size *N*, in terms of number of districts, was determined using a standard equation for cluster-based studies that measure a single percentage.

$$N = \left( \left[ 1 + (m - 1) \frac{k^2 p}{1 - p} \right] * 1.96^2 \frac{(1 - p)}{d^2 p} \right) / m$$

Here **p** stands for the primary outcome of interest, the percentage of TB underreporting (assumed to be 27%, based on a previous such study in adults), precision (**d**) selected to be 25%, a coefficient of variation (**k**) between 0.3–0.4 (based on a previous such study in adults), and cluster size (**m**) with harmonic mean between 33 and 77 (based on case notification data from BMU's across the country inflated by the estimated 2014 case detection rate), the study requires 12 districts to fulfil its objectives.

### Sample selection

The definition of a cluster in this study is the district, a well-defined geographical and administrative area, with low chances of broken linkages due to patients seeking health services from

**Table 1. Selection of districts based on probability proportional to population size.**

| Strata | Number of districts | Population (≤14 years) | Percentage share of national | Number of clusters | | |
| --- | --- | --- | --- | --- | --- | --- |
| | | | | *All get one* | *Additional** | *Total* |
| Balochistan | 30 | 2947212 | 5% | 1 | 0 | 1 |
| KPK | 25 | 8239283 | 14% | 1 | 1 | 2 |
| Punjab | 36 | 33027380 | 55% | 1 | 3 | 4 |
| Sindh | 23 | 13712676 | 23% | 1 | 2 | 3 |
| AJK | 10 | 1452889 | 2% | 1 | 0 | 1 |
| GB | 7 | 412616 | 1% | 1 | 0 | 1 |
| **National** | **131** | **59792055** | **100%** | **6** | **6** | **12** |

*Proportional to national share

the private sector in neighbouring districts. Sampling of 12 districts (clusters) out of 144 across the country was done, with probability proportional to population size of provinces and regions (Table 1). Prospective collection of data for cases diagnosed by all health-care providers within these districts for a specified time period, followed by record-linkage with an electronic, case-based national NTP database [6].

A total of 9,786 health facilities that were providing health services to presumptive child TB cases were mapped and invited to participate in the study. Mapping of all health facilities and health care providers was done through mobile based GPS system in the 12 selected districts/clusters. Public providers include the NTP and non-NTP public health services (public hospitals, university hospitals, medical organizations, social security, the army, etc.). Private health services include private hospitals, teaching hospitals and universities, clinics of GPs, health facilities run by NGOs, and informal health providers. Private laboratories were also included in this study.

## Case-definitions

A child TB case was aged less than 15 years, was diagnosed either pulmonary or extra pulmonary TB disease, and categorised either as a new case or retreatment. A child with presumptive TB was defined as someone with an unexplained cough for more than 2 weeks or/and at least one other TB symptom recommended by Pakistan Paediatric Association scoring chart [12].

**Bacteriologically-confirmed.** A bacteriologically confirmed TB case is one from whom a biological specimen is positive by smear microscopy, culture or rapid diagnostics like GeneXpert MTB/RIF. Data for all such cases was collected as part of study investigations, regardless of whether TB treatment has started. However, all diagnosed child TB cases were included in the analysis.

**Clinically-diagnosed.** A clinically diagnosed TB case has no bacteriological confirmation but has been diagnosed with active TB by a clinician or other medical practitioner. This definition also includes child TB cases diagnosed with X-ray abnormalities or suggestive histology and extra pulmonary cases with laboratory confirmation. Data for all diagnosed cases was collected regardless of whether TB treatment had started.

## Data collection tools and procedure

A health facility register for individuals with presumptive child TB cases was introduced in each NTP public and Non-NTP (public and private) health facilities in which all information regarding all such individuals and their management was recorded. This included full name of the patient, full contact address (with mobile number), age, sex, source of referral, number of specimens examined, results of the tests, final diagnosis and treating physician. In tertiary care hospitals, these registers were placed at all entry points of child TB patients, such as pediatric specialist, chest physician and medical officer. These forms were filled out by health providers who are diagnosing child TB cases in their ordinary work. All these health providers were asked to fill information, without any intervention to change their practice, for 3 months (1 quarter) during data collection from April—June 2016. A laboratory register was introduced to all laboratories to gather the information regarding the tests done for diagnosis and result of tests. All health care providers who consented to participate in the study were first trained to capture the required information in the registers, and also the laboratory personnel was trained. The data was collected through cell phones (using the application developed by "Zong"). Then all data was uploaded to the server. At central level, NTP, research department accessed the data on daily basis for monitoring and supervision.

## Monitoring and supervision

The health facilities were visited weekly by field officers, provincial coordinators, district tuberculosis coordinators and a district supervisor to ensure the quality (completeness, correctness) of data collected, to conduct cross-checking the status of registration at NTP, and to contact the unregistered cases in NTP registers to verify the diagnosis made by NTP & non-NTP providers. Any case with doubtful diagnosis (based on existing data from medical records) was captured as part of the study but classified according to study case definitions.

## Data quality audit (DQA)

To ensure the validity of data, every record was cross checked from the hard copies to remove inconsistencies between hard and soft records. The process of DQA for crosschecking registers was completed one quarter after the end of the study period.

## Data analysis and management

Three sources of information were used: I) standard NTP TB-registers; 2) Patient registers in private health facilities; 3) Patient registers in private laboratories. The study was based on two databases: health facility mapping and health facilities reporting. At first stage; de-duplication on both databases was done through probabilistic matching by comparing the following variables of each case: First name, last name, father's name, grandfather, district, sex, age, site of disease, hospital name, type of health facility. Then a manual review of multi-matches (one case matching to many) was done by research team to remove duplicates, this was done for 450 cases. Lacking unique identifiers of patients record linkage was carried out as outlined in guidance issued by WHO [6], by cross-checking the notifications of non-NTP facilities, compared with official public district TB registers, using the combination of the first name, father's name and family name in English language as identifiers. In case one of the names was missing other available identifiers were also compared. Presence or absence in the official district NTP register was recorded. To correct for possible misclassification (between notified and un-notified cases to NTP) the NTP registers were examined for the period between two quarters before and one quarter after the study period. Then databases were combined e.g. notification/ study dataset including matched and unique records for final analysis.

## Ethical considerations

Ethical clearance (registration # NBC 192 given in 2015) was granted from the Pakistan medical and Research Council before execution of the study as well as the WHO Ethics Committee for the East-Mediterranean region. Informed consent was taken from the NTP and non-NTP providers but not from the patients as this study was based on record review and no intervention was targeted on TB patients and ethics committee waived the need for informed consent from patients for their data to be used in research. However, patient identifiers were required for record linkage. Data was maintained in electronic register with a password-protected code and only the principal investigator or an authorized person had access to the data for analysis purposes.

## Results

A total of 7,125 presumptive child TB cases were detected by all participating health facilities between April—June, 2016, where 6,519 were diagnosed in the private health services, and 606 in private laboratories (Table 2). The age group 10-14y was largest (41%) and girls dominated (64%). From these children with presumptive TB, 5258 were diagnosed as TB and put on

**Table 2. Characteristics of children with presumptive tuberculosis in the study: Apr—Jun 2016.**

|  | Private Health Facility | Laboratory | Total |
|---|---|---|---|
|  | n (%) | n (%) |  |
| Age (in year) |  |  |  |
| <1 | 113 (1.7) | 2 (0.3) | 115 (1.6) |
| 1–4 | 1578 (24.2) | 61 (10.1) | 1639 (23.0) |
| 5–9 | 2240 (34.4) | 197 (32.5) | 2437 (34.2) |
| 10–14 | 2,588 (39.7) | 346 (57.1) | 2934 (41.2) |
| Gender |  |  |  |
| Male | 2,320 (35.6) | 220 (36.3) | 2540 (35.6) |
| Female | 4,199 (64.4) | 386 (63.7) | 4585 (64.4) |
| Total | 6519 | 606 | 7125 |

treatment (Table 3); in private health facilities 498 child TB cases were bacteriological positive while 4,695 child TB cases were clinically diagnosed; in laboratories only 65 were diagnosed bacteriological confirmed.

The record linkage indicated that 5193 (5006+187) child TB cases were identified by the private health providers and 65 (64+1) by the laboratories in the 12 study districts (Fig 1). Of these, 188 (187+1) were recorded in the NTP TB register, with an additional 1,267 cases registered only among the NTP sites. The level of underreporting nationally was 78%, with marked differences between provinces. The highest level of underreporting was found in Jhal Magsi, followed by Hafizabad and Pallundary (Table 4).

Table 5 shows the potential risk factors and uncertainty interval of underreporting. The underreporting was higher in boys 84% (78%–91%) than girls 68% (57%–79%). The underreporting was 76% (65%-87%) for bacteriologically confirmed TB cases and 78% (68%–88%) for clinically diagnosed cases.

**Table 3. Identified child TB cases by the private health providers & laboratories in selected districts of Pakistan: Apr—Jun, 2016.**

| Province | District | Health Facility | | Laboratory | Total |
|---|---|---|---|---|---|
|  |  | Bacteriological positive | Clinically Diagnosed | Bacteriological positive |  |
| Punjab | Attock | 31 | 411 | 1 | 443 |
|  | Chiniot | 108 | 229 | 1 | 338 |
|  | Hafizabad | 15 | 540 | 0 | 555 |
|  | Vehari | 30 | 303 | 0 | 333 |
| Sindh | Shikarpur | 8 | 386 | 0 | 394 |
|  | Hyderabad | 10 | 828 | 5 | 843 |
|  | Karachi | 130 | 812 | 26 | 968 |
| KPK | Buner | 11 | 137 | 0 | 148 |
|  | Peshawar | 126 | 873 | 32 | 1031 |
| AJK | Pallundary | 6 | 108 | 0 | 114 |
| Baloschistan | Jhal Magsi | 19 | 8 | 0 | 27 |
| GB | Ghizer | 4 | 60 | 0 | 64 |
| **Total** |  | **498** | 4695 | 65 | 5258 |

AJK = Azad Jammu & Kashmir, KPK = Khyber Pakhtunkhwa, GB = Gilgit Baltistan

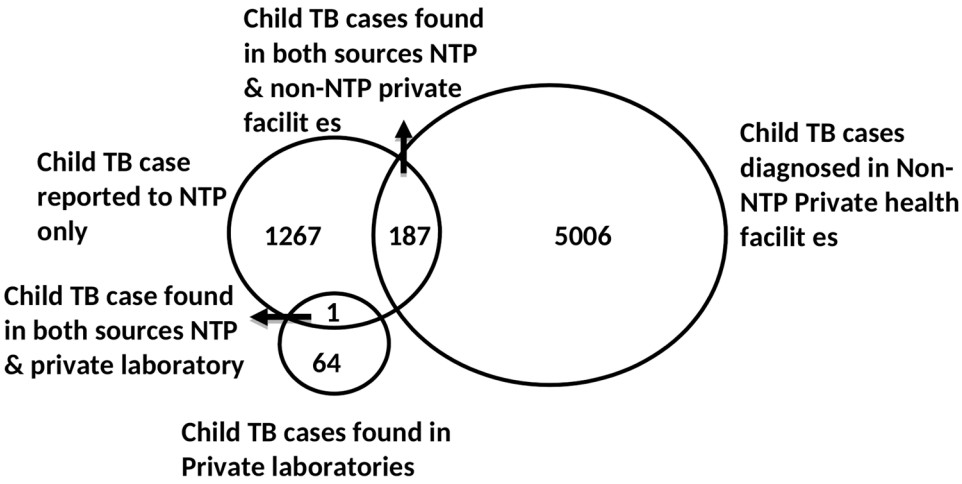

Level of under-report ng: 78

**Fig 1. Venn diagram showing child TB cases by source of identification.** NTP = National Tuberculosis Control Programme.

## Discussion

In this child TB inventory study in Pakistan we found that a substantial proportion of children diagnosed TB were detected by private providers, and the majority (89%) of child TB cases were clinically diagnosed. The proportion of notified child TB cases after record linkage was quite low and a massive underreporting (78%) was found. There was a marked difference between the provinces, and we notice that the highly populated Punjab contributed 64% of country TB case notification, and had a high level of underreporting. Proper training of private health providers and encouraging a "public-private-mix" approach to diagnosis and referral of

**Table 4. Child tuberculosis detected and notified in Pakistan, by district: Apr—Jun, 2016.**

| Districts | Child TB cases notified to NTP Only Q2 2016 | TB inventory study | | Underreporting |
|---|---|---|---|---|
| | | Cases reported to NTP | Cases not reported to NTP | |
| | (a) | (b) | (c) | [c/(a+b+c)] |
| Attock | 50 | 34 | 409 | 83% |
| Buner | 32 | 43 | 105 | 58% |
| Chiniot | 34 | 4 | 334 | 90% |
| Ghizer | 11 | 0 | 64 | 85% |
| Hafizabad | 5 | 12 | 543 | 97% |
| Hyderabad | 69 | 35 | 808 | 89% |
| Jhal maghsi | 0 | 0 | 27 | 100% |
| Karachi | 584 | 12 | 956 | 62% |
| Pallundary | 9 | 2 | 112 | 91% |
| Peshawar | 339 | 37 | 994 | 73% |
| Shikarpur | 54 | 2 | 392 | 88% |
| Vehari | 80 | 7 | 326 | 79% |
| Total | 1267 | 188 | 5070 | 78% |

**Table 5. Underreporting (overall and by risk factor) estimated taking into consideration the sampling design.**

| | Underreporting | |
|---|---|---|
| | **Best estimate** | **95% C.I** |
| Overall | 78% | 68%–87% |
| Gender | | |
| • Boys | 84% | 78%–91% |
| • Girls | 68% | 57%–79% |
| Case Type | | |
| • Bacteriological confirmed | 76% | 65%–87% |
| • Clinically diagnosed | 78% | 68%–88% |

C.I = confidence Interval

child TB cases to the National program for notifications could reduce the observed underreporting.

Underreporting has been described before, and a mathematical modelling study by Dodd in 22 high-burden countries predicted that the incidence of child TB is much higher than the number of notifications, and the global burden is perhaps 25% higher than the prediction for these countries Similar trends in under reporting of childhood TB cases have been reported elsewhere [13–15]. Effective household contact tracing through intensified case finding and engaging private providers who manage child TB could find more of these undetected or unreported cases.[16,17].

Factors associated with low case detection among children include poor record-keeping, lack of access to records on child TB cases for reporting. and inability to produce sputum for many children (only 30% of child TB cases can be detected through smear microscopy [16,18,19]. Other factors include limited capacity of healthcare providers to diagnose childhood TB and ineffective contact tracing [20,21]. A variety of factors have been identified as the leading causes of poor utilization of public health services, including poor socio-economic status, distance and accessibility to tertiary care level, cultural beliefs and perceptions, and low awareness and education.

During 2006–07, NTP Pakistan developed its national child TB guidelines in collaboration with Pakistan Paediatric Association [12]. Our study showed that these guidelines were underutilized in diagnosing child TB cases by the private providers across Pakistan. From other studies, it is evident that screening for child TB is feasible and cost effective using a symptom-based approach [22–24]. To improve the child TB case diagnostic practices of private health providers, wide dissemination of Pakistan Paediatric Association scoring guidelines, including chest-X-ray reading, needs to be properly followed and linked to further training of health care providers; this might strengthen the public-private mix approach. Its use should not be limited to tertiary care hospitals. The high proportion of children found in this study who were diagnosed with TB but not reported to the NTP confirms the potential significance of the non-NTP health providers. TB case notification is in principle mandatory, and private providers can be linked with NTP to increase the TB notification perhaps by reminding them that NTP will provide drugs free of cost. Contact tracing around index cases will find more cases of child TB and latent TB infection, and should be treated. There is also need to devise some strategies to increase utilization of GeneXpert MTB/RIF or develop some transportation referral mechanism across Pakistan especially in remote areas to improve the basis for diagnosis.

In our study, under-reporting was higher in males (84%) than females (68%). The Detected-to-notified case ratio was 70% higher in men than women. Other TB prevalence surveys in Asia have also shown higher such ratios for men than women [25].

Our study showed that a total of 4,695 (89%) child TB cases were diagnosed clinically, often based solely on clinical signs and chest X-ray. This may partly reflect the limited availability in remote areas of diagnostic investigations like GenXpert, TST (Tuberculin skin test) and Gastric Lavage. The same obstacles diagnosing child TB cases has been observed elsewhere [26,27]. A chest x-ray may give the clinicians a good indication that the patient suffers from TB, but bacteriological confirmation by rapid diagnostic tests to increase proportion of bacteriological confirmed child TB cases is highly recommended whenever available [28]. In children induced sputum, nasopharyngeal and gastric aspirates provide alternative respiratory samples in children who cannot expectorate. Fine needle aspiration biopsy could be an excellent option in children with peripheral lymphadenitis. Still, even such investigations some cases will still need to be diagnosed without clear evidence.

Suboptimal contact tracing and limited availability of appropriate diagnostic tools are big challenges in diagnoses of childhood TB in Pakistan. There is a need for better and sustainable public-private partnerships and robust electronic surveillance system to capture all child TB cases reported by strengthening child TB surveillance system, using current District Health Management system. Currently, child TB care services are confined to only tertiary care hospitals in Pakistan; so the services should be considered in other health facilities in public and private sectors, based on skills. Gastric lavage is often not available in many settings; sputum induction may be encouraged; sputum transportation system for Xpert/smear and proper referral linkages with peripheral centres should be ensured. In order to understand more about underreporting there is also need for research on child TB care delivery issues, like investigation practices, contact investigation and referrals, engagement of tertiary care hospitals and informal providers.

This study is as far as we know the first national inventory study in the world to assess underreporting of child TB cases. Major challenges in the study was a lengthy data collection tool, and many busy GPs didn't have time to fill themselves, so either the data collectors or clinic staff entered the data, enhanced by the use of electronic questionnaires by the data collectors. Also, we can not assess the size of a potential Hawthorne effect during the study, where facilities in the study were more likely to report cases than those not studied.

## Conclusion

This is the first national TB inventory study globally among children, and it has provided much needed data to improve our understanding of TB burden among this vulnerable subpopulation. The study confirmed that childhood TB is seriously underreported in Pakistan. TB surveillance in the country should be strengthened to address this collaborating with GPs and paediatricians.

## Acknowledgments

We are thankful to WHO headquarters and country office Pakistan for providing technical support for this study. We thank Kunju Shaji from Public Health England who supported record linkage. We are grateful to all survey field, central staff (research unit) and NTP management for continued support. We are thankful to all health care providers participated in this study. We highly acknowledge the corporation and facilitation of PTP Managers and PTOs from provisional TB control programs of Punjab, Sindh, Khyber Pakhtunkhwa, Balochistan, Azad Jammu & Kashmir and Gilgit Baltistan. University of Bergen provided a good academic background.

## Author Contributions

**Conceptualization:** Razia Fatima, Aashifa Yaqoob, Ejaz Qadeer, Charalambos Sismanidis.

**Data curation:** Aashifa Yaqoob, Charalambos Sismanidis.

**Formal analysis:** Aashifa Yaqoob, Charalambos Sismanidis.

**Funding acquisition:** Razia Fatima, Charalambos Sismanidis.

**Investigation:** Razia Fatima, Aashifa Yaqoob, Charalambos Sismanidis.

**Methodology:** Razia Fatima, Aashifa Yaqoob, Charalambos Sismanidis.

**Project administration:** Razia Fatima, Aashifa Yaqoob, Ejaz Qadeer.

**Resources:** Razia Fatima, Aashifa Yaqoob, Charalambos Sismanidis.

**Software:** Aashifa Yaqoob, Charalambos Sismanidis.

**Supervision:** Razia Fatima, Aashifa Yaqoob, Sven Gudmund Hinderaker.

**Validation:** Razia Fatima, Aashifa Yaqoob, Charalambos Sismanidis.

**Visualization:** Razia Fatima, Aashifa Yaqoob, Charalambos Sismanidis.

**Writing – original draft:** Razia Fatima, Aashifa Yaqoob.

**Writing – review & editing:** Razia Fatima, Aashifa Yaqoob, Ejaz Qadeer, Sven Gudmund Hinderaker, Aamer Ikram, Charalambos Sismanidis.

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
