## [Decision Letter · Decision Letter 0]

18 Jul 2019

PONE-D-19-14628

Measuring and addressing the childhood TB reporting gaps in Pakistan: the first ever national TB inventory study among children

PLOS ONE

Dear Dr. Yaqoob:

Thank you for submitting your manuscript to PLOS ONE. After careful consideration, we feel that this is an important paper which has merit, but does not fully meet PLOS ONE’s publication criteria as it currently stands. Therefore, we invite you to submit a revised version of the manuscript that addresses the points raised during the review process.

This is potentially a very important paper, which highights profound underreporting of tuberculosis rates in children in Pakistan. As such, the data reported in this manuscript have the potential to impact national health policies, guidelines, and recommended processes, in regards to diagnosis, reporting, and linkage for data related to pediatric TB cases.

To facilitate careful revision and rapid re-submission of this paper to PLOS ONE, the reviewers' comments have been collated into a single summary document, entitled "Editor's summary of suggested reviewer revisions for manuscript PONE-D-19-14628." This document is further divided into categories of "Major Revisions," "Minor Revisions," and "Discretionary Revisions." Please feel free to utilize this document, if desired, as an outline to structure the rebuttal letter which must accompany your re-submission.

Please note that, typically, all Major Revisions MUST be substantively addressed in the re-submission. Suggested Minor Revisions should typically also be addressed prior to re-submission -- however, if the co-authors have a very compelling justification for why this is not the case, then that justification should be outlined in the rebuttal letter which accompanies the re-submission.  Typically, Discretionary Revisions are not required to be addressed in order for a re-submitted manuscript to be further considered for publication. However, if is often wise for authors to carefully consider if such suggested discretionary revisions would offer additional clarity or context, and therefore, strengthen the manuscript.

We would appreciate receiving your revised manuscript by 16 August 2019. To enhance the reproducibility of your results, we recommend that if applicable you deposit your laboratory protocols in protocols.io, where a protocol can be assigned its own identifier (DOI) such that it can be cited independently in the future. For instructions see: http://journals.plos.org/plosone/s/submission-guidelines#loc-laboratory-protocols

We look forward to receiving your revised manuscript.

Kind regards,

Sherri L. Bucher, PhD, MA

Academic Editor

PLOS ONE

Journal Requirements:

The study was carried out with funding from UNITAID, channelled through the STEP-TB project that was led by the TB Alliance and WHO.

NO -The funders had no role in study design, data collection and analysis, decision to publish, or preparation of the manuscript.

3. In ethics statement in the manuscript and in the online submission form, please provide additional information about the patient records used in your study. Specifically, please ensure that you have discussed whether all data were fully anonymized before you accessed them and/or whether the IRB or ethics committee waived the requirement for informed consent.

5. Please include your tables as part of your main manuscript and remove the individual files. Please note that supplementary tables (should remain/ be uploaded) as separate "supporting information" files

Reviewers' comments:

Reviewer's Responses to Questions

**Comments to the Author**

1. Is the manuscript technically sound, and do the data support the conclusions?

Reviewer #1: Yes

Reviewer #2: No

Reviewer #3: Yes

Reviewer #4: Yes

2. Has the statistical analysis been performed appropriately and rigorously? 

Reviewer #1: I Don't Know

Reviewer #2: No

Reviewer #3: Yes

Reviewer #4: Yes

3. Have the authors made all data underlying the findings in their manuscript fully available?

Reviewer #1: No

Reviewer #2: No

Reviewer #3: Yes

Reviewer #4: Yes

4. Is the manuscript presented in an intelligible fashion and written in standard English?

Reviewer #1: Yes

Reviewer #2: No

Reviewer #3: Yes

Reviewer #4: Yes

5. Review Comments to the Author

Reviewer #1: The article focuses on underreporting of childhood tuberculosis, which is the urgent problem and should be emphasized as a publication in PLOSONE. This could be an important step to improve care in childhood TB in Pakistan in the future.

However, some points should be concerned

1. In 2017, Pakistan is a high burden TB country with estimated incidence (includes HIV+TB) of 525,000 cases. It is generally reported that at least 10%–15% of cases in the world and up to 25% of those arising in countries with high TB burden occur in children. The authors showed 5,258 childhood TB cases from the districts that sampled in a 3-month-period study. How many total childhood TB cases in the Pakistan reflected from this number?

2. How to select the 12 districts across the country? In page 5, “with probability proportional to population size provinces and regions” (Table 1)?? No detail of this in Table 1.

3. Underreporting was 78%; approximately 90% of TB cases are from non-NTP private health facilities. Should we discuss more why the NTP and public health facilities do not work?

4. The tests used for TB diagnosis is an important issue. The molecular test, such as GeneXpert is available and used countrywide or only in some cities. Of note, in Chiniot and Karachi district has high percentage of the cases that bacteriologically confirmed which is different from other districts even in the same provinces. Any explanation for this? Surprisingly, almost 30% in Chiniot was bacteriologically confirmed but very low reporting (90% in Table 3). So this is not the problem of uncertain TB diagnosis. The main reason of this underreporting should be also discussed.

5. Importantly, the authors should comment more on policies of the National Tuberculosis Control Program in Pakistan as well including routine reporting of childhood TB, BCG vaccine at birth, treatment of latent TB infection in household contact children, support of GeneXpert and reimbursement, that may bring better understanding to the reader.

Thank you

Reviewer #2: The approach to the analysis needs improvement/additional analysis to rule out mis-classification bias

Results tables are misleading in the way they are presented

The authors will need to improve on the structue of sentences to improve on clarity.

Additional comments are in the reviewer comments section

Reviewer #3: These investigators have documented significant underreporting of childhood TB cases in Pakistan. A prospective cross-sectional study design was used. Districts were utilized as the unit, with sampling done within each district. Study design was appropriate; all sectors potentially diagnosing childhood TB cases were instructed to complete forms with demographic and diagnostic information for all childhood TB diagnosed within the prospective study timeframe. They were instructed to not otherwise change any practices. Data from these prospective case reports were then compared to the National database for the same time period, with appropriate assumptions made for case linking between the two databases. Results demonstrate significant under-reporting of cases – nearly 80%. Males were more likely under-reported than females. Surprisingly bacteriologically diagnosed cases were under-reported at the same percentage as clinical diagnosis.

The discussion is appropriate but could be improved by suggestions to make reporting more robust. For instance could it be made mandatory to report a case when a chemist receives a prescription for treatment drugs?

Other minor comments:

1. The authors might want to comment on the possibility of a Hawthorne effect, where during the study facilities were more likely to report than usual. This might especially have occurred if the participants were aware of the study purpose.

2. The Venn diagram needs to be better explained. I assume that the overlaps are where a case appeared in both databases. However, the way it is labeled and the legend reads, it appears that the overlap are cases whose TB was diagnosed in both facilities. Please reword the legend to clarify.

3. The authors might consider use of a map utilizing the data in Table 3. The map could also illustrate where the non-surveyed tribal areas are located.

Reviewer #4: Kindly respond to some minor comments

Full title is not aligned with short title, kindly make corrections

Abstract should start with an "Introduction"

kindly rephrase following lines in the abstract"A nationwide, prospective cross sectional, cluster-sampled design was used" this is not a study design

Introduction section line 82, please give reference

Methodology: please give study design name

kindly mention the eligibility criteria for selection of study subjects

it is suggested to use "Sample selection" instead of sampling design.

In the manuscript, table 1 is "Characteristic of children" please make corrections

it would be interesting if details of "data collection through cell phone" is sharedd such as name of android device, name of android application" etc...

For data quality audit which of the records is taken as correct or gold standard?

it would be nice to discuss reasons for gaps in reporting or underreporting

in discussion section please give some information on 'periphery'

6. PLOS authors have the option to publish the peer review history of their article (what does this mean?). If published, this will include your full peer review and any attached files.

Reviewer #1: No

Reviewer #2: No

Reviewer #3: Yes: Edward A Liechty

Reviewer #4: Yes: Sarah Saleem

---

## [Author Response · Author response to Decision Letter 0]

1 Nov 2019

Reply: Plose One style template is followed as given above 

The study was carried out with funding from UNITAID, channelled through the STEP-TB project that was led by the TB Alliance and WHO.

NO -The funders had no role in study design, data collection and analysis, decision to publish, or preparation of the manuscript.

Reply: Please update Funding statement as below 

“The study was carried out with funding from UNITAID, channelled through the STEP-TB project that was led by the TB Alliance and WHO. The funders had no role in study design, data collection and analysis, decision to publish, or preparation of the manuscript.”

3. In ethics statement in the manuscript and in the online submission form, please provide additional information about the patient records used in your study. Specifically, please ensure that you have discussed whether all data were fully anonymized before you accessed them and/or whether the IRB or ethics committee waived the requirement for informed consent.

Reply: ethical approval was obtained from National Bioethics Committee of “Pakistan medical and Research Council” ensuring all ethics concerns related to the study

Reply: Special permission for program is required for data availability. We can provide it on request because of office policy restriction for data sharing. Data can be accessed through “drrazia@ntp.gov.pk” or “aashifa.yaqoob@ntp.gov.pk” by contacting IRC ethics committee, Common management unit (HIV/AIDS, TB & Malaria) to which data requests can be sent.

Reply: All above points are incorporated in cover letter

5. Please include your tables as part of your main manuscript and remove the individual files. Please note that supplementary tables (should remain/ be uploaded) as separate "supporting information" files

Reply: Table is included in manuscript

---

## [Editor Report · Decision Letter 1]

16 Dec 2019

Measuring and addressing the childhood tuberculosis reporting gaps in Pakistan: the first ever national inventory study among children

PONE-D-19-14628R1

Dear Dr. Yaqoob,

We are pleased to inform you that your manuscript has been judged scientifically suitable for publication and will be formally accepted for publication once it complies with all outstanding technical requirements.

With kind regards,

Sherri Lynn Bucher, PhD

Guest Editor

PLOS ONE

Additional Editor Comments (optional):

Dear Dr. Yaqoob--

Thank you for submission of the revised manuscript, and for being responsive to the revisions requested by the reviewers. It is recommended that this manuscript be accepted for publication in PLoS ONE. Congratulations!

Best,

Dr. Sherri Bucher
---

## [Editor Report · Acceptance letter]

18 Dec 2019

PONE-D-19-14628R1 

Measuring and addressing the childhood tuberculosis reporting gaps in Pakistan: the first ever national inventory study among children 

Dear Dr. Yaqoob:

I am pleased to inform you that your manuscript has been deemed suitable for publication in PLOS ONE. Congratulations! Your manuscript is now with our production department. 

With kind regards,

on behalf of

Dr. Sherri Lynn Bucher 

Guest Editor

PLOS ONE